# Determinants of WHO recommended COVID-19 prevention measures among pregnant women attending antenatal care during the third wave of COVID-19 in eastern Ethiopia, 2021

**Astawus Alemayehu** [1] *, **Mohammed Yusuf**[2], **Abebaw Demissie**[3], **Yasin Abdullahi**[4], **Lemessa Oljira**[5], **Nega Assefa**[6]

1 Department of Public Health, Harar Health Science College, Harar, Ethiopia, 2 Department of Nursing, Harar Health Science College, Harar, Ethiopia, 3 Department of Anesthesia, Harar Health Science College, Harar, Ethiopia, 4 Department of Management, Harar Health Science College, Harar, Ethiopia, 5 School of Public Health, College of Health and Medical Sciences, Haramaya University, Harar, Ethiopia, 6 College of Health and Medical Sciences, Haramaya University, Harar, Ethiopia

* astawusalemayehu@gmail.com

## Abstract

### Background

The novel coronavirus disease has emerged as the most pressing global health issue. In women with COVID-19 disease, pregnancy confers a substantial additional risk of morbidity and mortality.

### Objective

This study aimed to assess WHO-recommended COVID-19 prevention practices and determinant factors among pregnant women attending antenatal care during the third wave of COVID-19 in eastern Ethiopia.

### Methods

An Institutional-based cross-sectional study was conducted among 422 pregnant women attending antenatal care in Harar, from October 10 to November 10, 2021. The sample size was proportionally allocated to all healthcare facilities, then the study participants were selected using systematic random sampling. Descriptive summary statistics were done. Logistic regression analyses were computed to identify associations between dependent and independent variables. Variables with a p-value < 0.05 were declared statistically significant.

### Result

Out of 422 pregnant women, 61.6% of them had good WHO Recommended COVID-19 Prevention Practices. Those with age 25–34 years (AOR: 9.7, 95%CI: 4.8, 19.3), age 35–44 years (AOR:4.8, 95%CI: 2.6, 9.03), monthly income > 10,000 ETB (AOR: 9.4, 95%CI: 2.1,

**Data Availability Statement:** All relevant data are within the manuscript and uploaded as Supporting information files.

**Funding:** This study was funded by Harar Health Science College. funders plays a role in the study design, data collection and analysis, decision to publish, or preparation of the manuscript.

**Competing interests:** The authors have declared that no competing interests exist.

**Abbreviations:** CDC, Communicable Disease Control; FMoH, Federal Ministry of Health; IHREC, Institutional Health Research Ethics Committee; MERS, Middle East Respiratory Syndrome; SARS Cov-2, Severe Acute Respiratory Syndrome Corona-virus 2; SARS, Severe Acute Respiratory Syndrome; SPSS, Statistical Package for Social Science; WHO, World Health Organization.

42.1), being a student (AOR: 10, 95%CI: 2.3, 47.1), having a good level of knowledge (AOR: 2.3, 95%CI:1.4, 3.8), and having ≥10 family members (AOR: 0.24, 95%CI: 0.06, 0.9) were found to have a significant association with WHO recommended prevention practice among pregnant women.

## Conclusion

Overall, the WHO-recommended COVID-19 prevention practice among pregnant women attending antenatal care was good, but it needs improvement. In order to improve prevention practices among pregnant women, Harari Regional Health Bureau and other stakeholders should provide repeated, targeted, and tailored information to pregnant women and the community at large through different media.

## Introduction

Coronavirus disease (COVID-19) has emerged as the most pressing global health issue [1]. The first cases of COVID-19 infection were identified in Wuhan, China, in December 2019 [2, 3]. Since then, the virus has spread to almost every nation in the world and was declared a pandemic by the World Health Organization on March 11, 2020 [4]. In Ethiopia, the Federal Ministry of Health (FMoH) confirmed the first case of COVID-19 in Addis Ababa on March 13, 2020 [5]. Globally, as of October 18 of 2021, there had been more than 200 million and 4 million COVID-19 confirmed cases and deaths, respectively. In Africa, there have been more than 6 million confirmed COVID-19 cases, with 149,041 deaths [6]. Between 3 January 2020 and 18 October 2021, there were more than 300 thousand confirmed cases of COVID-19 in Ethiopia, with 6,197 deaths [7].

Several countries have seen a two-wave pattern of reported cases of COVID-19, the first and second wave [8]. The first wave of the pandemic hit a lot of countries hard and many patients died. The second wave pandemic is likewise a serious threat to society, with a huge toll in terms of human deaths and a catastrophic economic effect [9, 10]. In some countries, much higher infection numbers and more deaths were seen during the second wave [11]. On the other hand, wealthy countries have drastically reduced their mortality during their second wave [12]. However, patients in the second wave were younger, more pregnant, and post-partem [8].

Studies suggested that the majority of infections with COVID-19 cause a mild form of infection. However, older adults and people with comorbidities, including cardiovascular, respiratory diseases, and diabetes are at increased risk of severe illness and death, with men potentially at higher risk than women [13, 14].

In women with COVID-19 disease, pregnancy confers a substantial additional risk of morbidity [15]. Recent studies indicated that COVID-19 has been linked to a greater risk of serious illness and mortality in pregnant women than in non-pregnant women [16]. Pregnant women who have COVID-19 are also more likely to experience preterm delivery, fetal distress, low birth weight, and other negative pregnancy outcomes [17, 18].

The second wave of the COVID-19 pandemic was more severe than the first among pregnant women in terms of illness severity, intensive care unit hospitalization and invasive ventilation, and maternal death [19]. Similarly, in Spain, during the first wave, the severity of COVID-19 infection in pregnant women was modest [8], but during the second wave, the number of pregnant women hospitalized owing to COVID-19 infection increased ten-fold [20, 21].

Nations across the world have launched various COVID-19 prevention measures, including restricted movement, quarantine, and nationwide lockdowns [22]. Additionally, individual and community actions to improve hand hygiene, physical distancing, and the use of face masks including vaccination campaigns were also implemented [23, 24]. The government of Ethiopia has engaged in prevention activities including media campaigns to disseminate information on WHO-recommended preventive measures to the general population including pregnant women [24, 25]. Despite the implementation of such measures, the burden of the pandemic has not been reduced significantly.

As pregnant women are considered to be a risk population group there is a need to assess their prevention practice. So far, studies conducted among pregnant women in Ethiopia have focused only on assessing knowledge, attitude, and practice toward COVID-19 infection. There are only two studies that were conducted in northern and southern Ethiopia, that focused on pregnant women's practice on WHO-recommended COVID-19 prevention practice, which reported low COVID-19 preventive practice and knowledge [26, 27]. In addition, there was no prior study conducted on this specific topic in the Eastern part of Ethiopia. Therefore, this study has assessed WHO Recommended COVID-19 Prevention Practices and Determinant Factors Among Pregnant Women Attending Antenatal Care during the third wave of COVID-19 in Eastern Ethiopia.

## Materials and method

### Study setting

The study was carried out in Harari regional state from October 10 to November 10, 2021. Harar is located 526 kilometers away from the capital, Addis Ababa, and covers 334 square kilometers. It has a total population of 246,000 people, with over 60% of the population living in urban areas. In the Harari regional state currently, 14 health facilities provide antenatal care. These were two public hospitals (Jugal general hospital and Hiwot Fana Comprehensive Specialized University Hospital), seven health centers (Arategna, Aboker, Amir Nur, Jinela, Harawe, Hasengey, and Sofi), and five private clinics within the study area. All 14 health facilities in Harari regional state were included in the study.

### Study design

An institutional-based cross-sectional study design was utilized.

### Study population

All pregnant women who were attending antenatal care at healthcare facilities in Harar were our source population. Pregnant women who were attending antenatal care at all healthcare facilities during the data collection period were our study population.

### Eligibility criteria

All pregnant women who were attending antenatal care and had the willingness to participate in the study were included. Pregnant women who were unable to communicate or seriously ill were excluded from the study.

### Sample size determination

The sample size was determined by single population proportion with assumptions of proportion = 50%, with a 5% margin of error, and 95% confidence interval. Finally, the required

calculated sample size considering 10% (38) allowance for a non-response rate was 422.

$$n = \frac{Z\alpha^2\, p(1-p)}{d^2} = n = \frac{(1.96)^2\, 0.5(1-0.5)}{0.05^2} = 384, \text{ adding } 10\% \ (38) \approx 422$$

## Sampling procedure

We included all healthcare facilities currently providing antenatal care services for pregnant women in the Harari region. The sample size was proportionally allocated to each healthcare facility based on the number of pregnant women attending ANC. Then, a systematic random sampling technique was used to select the representative pregnant women using the $k^{th}$ interval 0.4178 = ~ 1which means every other pregnant woman was selected (Fig 1).

## Data collection tool and procedure

A face-to-face interview was conducted for data collection using a structured questionnaire. The questionnaire was prepared in the English language and was adapted after reviewing different relevant literature [26, 28], and WHO COVID-19 recommended prevention practice guidelines [29]. Then it was translated into the local language (Amharic and Afan Oromo). The questionnaire consisted of three parts; sociodemographic variables (age, residence, monthly income, educational status of women, educational status of the husband, family size, Occupation, age at first marriage, number of children, parity, and gravidity), Knowledge toward Prevention of COVID-19 variables and WHO COVID-19 recommended prevention practice questions were our outcome variables. For all knowledge, and practice questions participants who answered the "correct answer" or "Yes" were earned a "1 score", whereas participants who gave the "wrong answer" or "No" were earned a "0 score". Data were collected by five $4^{th}$ year nursing students.

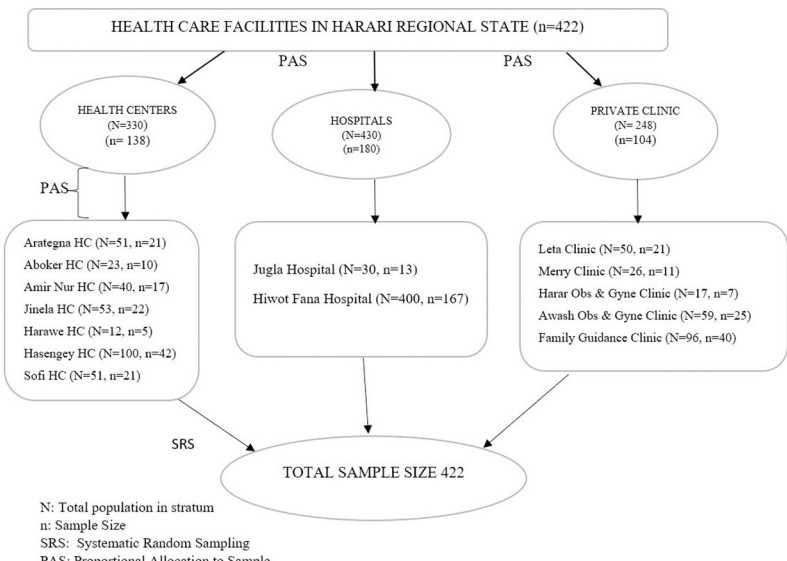

**Fig 1. Schematic presentation of sample technique and sample allocated for each health care facility in Harari Regional state, 2021.**

## Ethics approval

The protocol of this study for the subject recruitment process and participation in the study adhered to the Declaration of Helsinki's guidelines and an ethical approval letter was obtained from Harar Health Science College Institutional Health Research Ethics Committee (IHREC) with reference No. IHREC 2/10/9/21/2/14. Oral informed consent was obtained from participants before collecting data. This is because written consent was not feasible as some of the participants may be illiterate, and also for those participants under the age of 18, we obtained consent from their parents, during the study who came to health facilities with pregnant women. Then, the data collector asks the interviewee whether they are willing to participate in a study before each interview. If the participant agrees to participate, the data collector marks "Yes" in the informed consent form and signs his signature. The data collector can also mark "No" if the person refuses to participate. This form of obtaining consent was approved by the IHREC. All participants provided their consent prior to participating in the study. Participation was completely voluntary, and participants were free to withdraw from the study at any time without any consequence. Confidentiality of all information has been maintained.

## Data quality control

To assure the quality of the data, a three-day training was given for data collectors on how to interview and collect data. A pretest was done on 5% of the questionnaire on police hospital. For the questions or items found to be unclear, the required modifications were made accordingly. The reliability of the questionnaire was calculated by using SPSS v.26 and the value of Cronbach's alpha was found to be 0.897. Close supervision of the data collectors was made by the authors. Collected data was checked both in the field and at the end of each day after data collection, before data entry, for completeness and missing values. Double data entry was performed.

## Data processing and analysis

After the data was collected it was checked for completeness, clarity, and consistency. Then it was coded and entered into Epidata v.3.0, and for analysis transported to SPSS v.26. Descriptive summary statistics were produced in the form of percentages, mean, and standard deviation. Binary logistic regression analysis was computed to assess the association between the dependent and independent variables. In bivariate analysis, those variables with a p-value < 0.25 were moved to the multivariate analysis to control the effect of confounding variables. Finally, those variables in the multivariate analysis with a p-value < 0.05 were declared as having a statistically significant association. This study methodologically adheres to the STROBE checklist for cross-sectional study design for writing the finding of the study.

## Operational definition

**Level of knowledge.** When participants scored above the mean score from 7 knowledge questions it is declared as having good knowledge and when participants scored below the mean score it is declared as having poor knowledge [30].

**Level of practice.** When participants scored above the mean score from 7 practice questions it is declared as having good practice and when participants scored below the mean score it is declared as having poor practice [31, 32].

# Result

## Sociodemographic characteristics of pregnant women

A total of 422 pregnant women attending ANC participated in this study with a 100% response rate. The mean age of the participants was 31 (± 6.8). Among the participants, 32.9% and 39.1% were housewives and merchants, respectively. The majority, 79.1% of mothers had < 5 children (Table 1).

## Knowledge of pregnant women toward WHO recommended COVID-19 prevention

The majority, 89.8% of study participants knew washing hands for 20 seconds can prevent the virus, among these 50% and 39.8% live in urban and rural, respectively. Similarly, the vast majority, 94.3% of them knew that wearing a face mask can prevent virus transmission (Table 2).

**Table 1. Sociodemographic characteristics of pregnant women attending ANC in Harar, Eastern Ethiopia, 2021.**

| Sociodemographic characteristics of women | | Frequency | Percentage |
|---|---|---|---|
| Age in years | 15–24 | 95 | 22.5% |
| | 25–34 | 135 | 32.0% |
| | 35–44 | 192 | 45.5% |
| Residence | Urban | 237 | 56.2% |
| | Rural | 185 | 43.8% |
| Average monthly income | < 5000 ETB | 249 | 59.0% |
| | 5000–10,000 ETB | 152 | 36.0% |
| | ≥10,000 ETB | 21 | 5.0% |
| Educational status of women | Unable to read and write | 40 | 9.5% |
| | Primary education | 152 | 36.0% |
| | Secondary education | 170 | 40.3% |
| | Above secondary school | 60 | 14.2% |
| Educational status of Husband | Unable to read and write | 11 | 2.6% |
| | Primary education | 93 | 22.0% |
| | Secondary education | 159 | 37.7% |
| | Above secondary school | 159 | 37.7% |
| Family size | < 5 | 246 | 58.3% |
| | 5–9 | 156 | 37.0% |
| | ≥ 10 | 20 | 4.7% |
| Occupation | Student | 24 | 5.7% |
| | Housewife | 139 | 32.9% |
| | Merchant | 165 | 39.1% |
| | Civil servant | 94 | 22.3% |
| Age at first marriage | < 18 years | 95 | 22.5% |
| | ≥ 18 years | 327 | 77.5% |
| Number of Children | Have no child | 53 | 12.6% |
| | < 5 children | 334 | 79.1% |
| | 5–9 children | 23 | 5.5% |
| | ≥ 10 children | 12 | 2.8% |
| Number of pregnancies | First time | 175 | 41.5% |
| | 2–4 times | 212 | 50.2% |
| | ≥ 5 times | 35 | 8.3% |

**Table 2. Knowledge towards WHO recommended COVID-19 prevention among pregnant women attending antenatal care in Harar, Eastern Ethiopia, 2021.**

| Knowledge toward Covid-19 prevention | Urban n = 237 | Rural n = 185 | Total N = 422 | Percentage |
|---|---|---|---|---|
| Washing hands for 20 seconds can prevent the virus | | | | |
| Yes | 211 (50.0%) | 168 (39.8%) | 379 | 89.8% |
| No | 26 (6.2%) | 17 (4.0%) | 43 | 10.2% |
| Sneezing or coughing into arm/elbow can prevent the spread of the virus | | | | |
| Yes | 217 (51.4%) | 154 (36.5%) | 371 | 87.9% |
| No | 20 (4.7%) | 31 (7.3%) | 51 | 12.1% |
| Virus can be transmitted by shaking hands | | | | |
| Yes | 202 (47.9%) | 168 (39.8%) | 370 | 87.7% |
| No | 35 (8.3%) | 17 (4.0%) | 52 | 12.3% |
| Maintaining safe distance at least one meter can protect from the virus | | | | |
| Yes | 198 (46.9%) | 160 (37.9%) | 358 | 84.8% |
| No | 39 (9.2%) | 25 (5.9%) | 64 | 15.2% |
| Touching face can transfer the virus | | | | |
| Yes | 201 (47.6%) | 157 (37.2%) | 358 | 84.8% |
| No | 36 (8.5%) | 28 (6.6%) | 64 | 15.2% |
| Staying at Home can decrease the chance of getting infected | | | | |
| Yes | 188 (44.5%) | 143 (33.9%) | 331 | 78.4% |
| No | 49 (11.6%) | 42 (9.9%) | 91 | 21.6% |
| Wearing the mask can prevent the virus | | | | |
| Yes | 220 (52.1%) | 178 (42.2%) | 398 | 94.3% |
| No | 17 (4.0%) | 7 (1.6%) | 24 | 5.7% |

N = total sample size, n = sub group sample

## Practice of pregnant women toward WHO recommended COVID-19 prevention

Overall, 61.6% of pregnant women had good level of WHO recommended COVID-19 prevention practice (Fig 2). Of these, 53.5% of them live in urban areas and 48.5% are found between the ages of 35–44 years.

Out of the total 422, the majority, 82.7% and 72.7% of pregnant women wear face masks and wash their hands for 20 seconds, respectively. On the other hand, 43.1% of participants didn't avoid handshaking (Table 3).

## Factors associated with WHO recommended COVID-19 prevention practice

Women in the age group of 25–34 years were 9.7 (AOR: 9.7, 95%CI: 4.8, 19.3) times, and those with age 35–44 years were 4.8 (AOR:4.8, 95%CI: 2.6, 9.03) times more likely to practice WHO recommended COVID-19 prevention measures than those women in the age group of 15–24 years.

Women who had ≥10,000 ETB average monthly income were 9.4 (AOR: 9.4, 95%CI: 2.1, 42.1) times more likely to practice WHO recommended COVID-19 prevention measures than those women who had < 5000 ETB monthly income.

Pregnant women who are students were 10 (AOR: 10, 95%CI: 2.3, 47.1) times more likely to practice WHO-recommended COVID-19 prevention measures than civil servant pregnant women (Table 4).

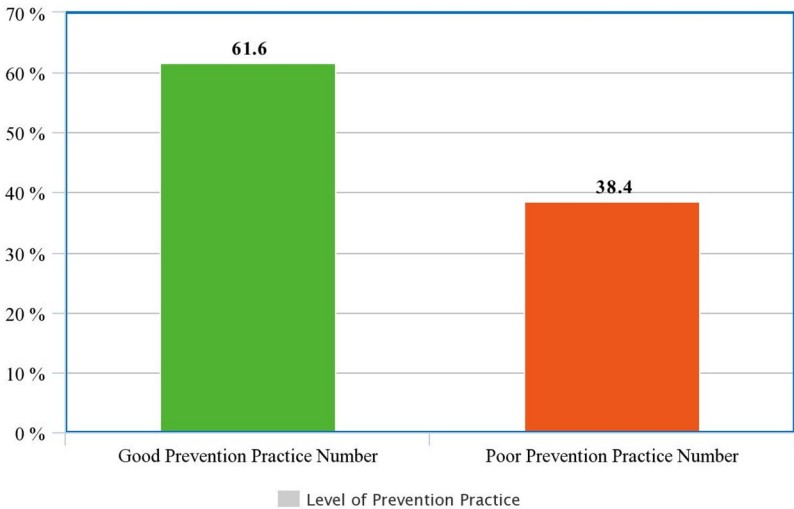

**Fig 2. Level of prevention practice among pregnant women toward WHO recommended COVID-19 prevention in Eastern Ethiopia, 2021.**

## Discussion

This study was the first study conducted in the eastern part of Ethiopia that tried to assess the level of prevention practice and determinant factors toward WHO-recommended COVID-19 prevention measures among pregnant women attending antenatal care.

**Table 3. WHO recommended COVID-19 prevention practice among pregnant women attending antenatal care in Harar, Eastern Ethiopia, 2021.**

| Practice toward Covid-19 prevention | Urban n = 237 | Rural n = 185 | Total N = 422 | Percentage |
|---|---|---|---|---|
| Washing a hand for 20 second | | | | |
| Yes | 173 (40.9%) | 134 (31.7%) | 307 | 72.7% |
| No | 64 (15.2%) | 51 (12.1%) | 115 | 27.3% |
| Sneezing/coughing into arm/elbow | | | | |
| Yes | 181 (42.9%) | 130 (30.8%) | 311 | 73.7% |
| No | 56 (13.3%) | 55 (13.0%) | 111 | 26.3% |
| Avoid shaking hands | | | | |
| Yes | 135 (32.0%) | 105 (24.9%) | 240 | 56.9% |
| No | 102 (24.2%) | 80 (19.0%) | 182 | 43.1% |
| Maintain a social distance of at least one meter | | | | |
| Yes | 139 (32.9%) | 112 (26.5%) | 251 | 59.5% |
| No | 98 (23.2%) | 73 (17.3%) | 171 | 40.5% |
| Avoid touching face | | | | |
| Yes | 123 (29.1%) | 119 (28.2%) | 242 | 57.3% |
| No | 114 (27.0%) | 66 (15.6%) | 180 | 42.7% |
| Staying at home quite often | | | | |
| Yes | 107 (25.3%) | 80 (18.6%) | 187 | 44.3% |
| No | 130 (30.8%) | 105 (24.9%) | 235 | 55.7% |
| Use face mask | | | | |
| Yes | 194 (46.0%) | 155 (36.7%) | 349 | 82.7% |
| No | 43 (10.2%) | 30 (7.1%) | 73 | 17.3% |

N = total sample size, n = subgroup sample

**Table 4. Bivariate and multivariate analysis for factors associated with WHO recommended COVID-19 prevention practice among pregnant women attending antenatal care in Harar, Eastern Ethiopia, 2021.**

| Factor variables | Practice toward Covid-19 Prevention | | COR CI: 95% | P-value | AOR CI: 95% | P-value |
|---|---|---|---|---|---|---|
| | Good | Poor | | | | |
| Age in years | | | | | | |
| 15–24 | 30 (31.6%) | 65 (68.4%) | 1 | | | |
| 25–34 | 104 (77.0%) | 31 (23.0%) | 7.2 (4.0, 13.1) | 0.0001 | 9.7 (4.8, 19.3) | **<0.001***  |
| 35–44 | 126 (65.6%) | 66 (34.4%) | 4.1 (2.4, 6.9) | 0.0001 | 4.8 (2.6, 9) | **<0.001***  |
| Residence | | | | | | |
| Urban | 139 (58.6%) | 98 (41.3%) | 0.75 (0.5, 1.1) | 0.157 | 1.6 (0.9, 2.7) | 0.066 |
| Rural | 121 (65.4%) | 64 (34.6%) | 1 | | 1 | |
| Monthly income | | | | | | |
| < 5000 ETB | 126 (50.6%) | 123(49.4%) | 1 | | | |
| 5000–10000 ETB | 117 (77.0%) | 35 (23.0%) | 3.2 (2.1, 5.1) | 0.0001 | 3.9 (2.1, 7.8) | **<0.001***  |
| ≥10,000 ETB | 17 (80.1%) | 4 (19.0%) | 4.1 (1.4, 12.6) | 0.013 | 9.4 (2.1, 42.1) | **<0.001***  |
| Education of women | | | | | | |
| Unable to read and write | 29 (72.5%) | 11 (27.5%) | 1 | | | |
| Primary education | 93 (61.2%) | 59 (38.8%) | 0.59 (0.3, 1.3) | 0.19 | 0.76 (0.3, 2.1) | 0.588 |
| Secondary education | 108 (63.5%) | 62 (36.5%) | 0.66 (0.3, 1.4) | 0.286 | 1.38 (0.4, 4.1) | 0.560 |
| Above secondary school | 30 (50.0%) | 30 (50.0%) | 0.38 (0.2, 0.9) | 0.027 | 1.48 (0.3, 6.1) | 0.585 |
| Education of Husband | | | | | | |
| Unable to read and write | 10 (91.0%) | 1 (9.0%) | 1 | | | |
| Primary education | 63 (67.7%) | 30 (32.3%) | 0.21 (0.03, 1.7) | 0.145 | 0.14 (0.1, 1.5) | 0.102 |
| Secondary education | 106 (66.7%) | 53 (33.3%) | 0.2 (0.25, 1.6) | 0.130 | 0.31 (0.3, 3.4) | 0.339 |
| Above secondary school | 81 (51.0%) | 78 (49.0%) | 0.10 (0.01, 0.8) | 0.033 | 0.12 (0.0, 1.5) | 0.098 |
| Family size | | | | | | |
| < 5 | 153 (62.2%) | 93 (37.8%) | 1 | | | |
| ≥5 | 107 (60.8%) | 69 (39.2%) | 1.1 (0.7, 1.6) | 0.77 | 0.89 (0.5, 1.6) | 0.715 |
| Occupation | | | | | | |
| Student | 19 (79.2%) | 5 (20.8%) | 3.3 (1.1, 9.7) | 0.026 | 10 (2.3, 47.1) | **0.003***  |
| Housewife | 94 (67.6%) | 45 (32.4%) | 1.8 (1.1, 3.1) | 0.027 | 2.9 (1.3, 6.6) | **0.009***  |
| Merchant | 97 (58.8%) | 68 (41.2%) | 1.3 (0.7, 2.1) | 0.382 | 1.4 (0.6, 3.1) | 0.410 |
| Civil servant | 50 (53.2%) | 44 (46.8%) | 1 | | | |
| Age at first marriage | | | | | | |
| < 18 years | 58 (61.0%) | 37 (39.0%) | 1.1 (0.6, 1.6) | 0.899 | 1.2 (0.7, 2.3) | 0.486 |
| ≥ 18 years | 202 (61.8%) | 125(38.2%) | 1 | | | |
| Number of children | | | | | | |
| Have no child | 35 (66.0%) | 18 (34.0%) | 1 | | | |
| < 5 children | 206 (61.7%) | 128(38.3%) | 0.82 (0.4, 1.5) | 0.543 | 2.1 (0.8, 5.1) | 0.113 |
| 5–9 children | 10 (43.5%) | 13 (56.5%) | 0.39 (0.14, 1.1) | 0.07 | 0.58 (0.1, 2.5) | 0.471 |
| ≥ 10 children | 9 (75.0%) | 3 (25.0%) | 1.5 (0.4, 6.4) | 0.551 | 3.2 (0.5, 21.8) | 0.243 |
| Number of pregnancies | | | | | | |
| First time | 113 (64.6%) | 62 (35.4%) | 1.5 (0.7, 3.2) | 0.252 | 1.5 (0.9, 2.8) | 0.213 |
| 2–4 times | 128 (60.4%) | 84 (39.6%) | 1.2 (0.6, 2.6) | 0.497 | 1.4 (0.7, 2.1) | 0.326 |
| ≥ 5 times | 19 (54.3%) | 16 (45.7%) | 1 | | | |
| Level of knowledge | | | | | | |
| Good knowledge | 163 (70.0%) | 69 (30.0%) | 2.3 (1.5, 3.4) | 0.0001 | 2.3 (1.4, 3.8) | **0.001***  |
| Poor knowledge | 97 (51.0%) | 93 (49.0%) | 1 | | | |

Note: Bold*, p-value <0.05 significant

Abbreviation: COR, Crude odds ratio; CI, Confidence interval; AOR, Adjusted odds ratio

In this study, most, 45.5%, of pregnant women ranged between the age group of 35–44 years with a mean age of 31(±6.8 SD). This is inconsistent with the studies conducted in Gurage zone [26], Debre Birhan [27], and Debre Tabor [33], Ethiopia, which reported the majority of pregnant women were found in the age group of 25–34 years, 47.4% with a mean age of 27.1 (± 5.2SD) 48.8% with a mean age of 27.2 (± 5.05SD), and 56.9% with a mean age of 27.1 (± 4.7SD), respectively. This variation could be due to differences in the age of marriage among women in the regions, sampled population and socio-culture.

The finding of this study indicates that 54.9% of pregnant women had good knowledge of COVID-19 prevention measures. This is similar to the finding reported from studies conducted in Gurage [26] and Gondar [34], Ethiopia, were 54.8% and 55% of pregnant women had good knowledge, respectively. On the other hand, our finding was higher than the study conducted in Debre Tabor [33] which reported 46.8% of participants had good knowledge. But it was much lower than the study conducted in Debre Birhan [27] (70.5%), and Ghana [35] (85.6%). This variation might be due to the difference in population, study area, and access to information.

The level of practice of pregnant women in this study toward Covid 19 prevention was 61.6%. this is higher than studies conducted in different areas, 46.6% in Ghana, west Africa [35]; 47.4%, 47.6%, and 56.1%, in Gondar [34], Debre Tabor [33], and Debre Birhan [27], Ethiopia, respectively. But lower than the finding from the study in Gurage zone [26], where 76.2% of the pregnant women had good practices toward Covid 19 prevention. The possible explanation for this observed variation could be due to differences in sampling population, study setting, and access to information through different media (TV, Radio, Facebook, and magazine).

Women in the 25–34 years age group were 9 times more likely to practice recommended COVID-19 prevention practices than those women in the 15–24 years age group. This is consistent with other similar studies conducted in different areas that indicated as the age of pregnant women increases the level of prevention practice also increases [26, 27, 35]. This might be due to the fact that women with increased age feel more responsible to protect themselves as well as their families and to exercise the recommended COVID-19 prevention practices.

Women with good knowledge about WHO-recommended COVID-19 prevention measures were 2 times more likely to exercise prevention practices than their counterparts. This finding agrees with the findings reported from Debre Tabor [33], Gondar [34], and Debre Birhan [27], Ethiopia. This might be due to the fact that access to information through different media (TV, Radio, Facebook, and magazine) and the provision of health information by healthcare facilities.

Our study revealed a new finding, that is, pregnant women with 5000–10000 ETB and > 10000 ETB monthly average income were 4 and 9 times more likely to exercise prevention practice than those who earn < 5000 ETB, respectively. This could be due to the fact that women with higher incomes have more access to have COVID-19 prevention items such as; face masks and hand sanitizers.

Regarding occupation, pregnant women who are students were 10 times more likely to exercise COVID-19 prevention practices than civil servants. This could be due to more strict regulations imposed by schools on students to comply with Covid 19 prevention measures.

Similarly, housewives were 3 times more likely to have good prevention practices than civil servants. This could be due to the fact that housewives usually stay at home and had a lower chance of contact with others than civil servants.

## Limitations of the study

The study used a cross-sectional study design. Therefore, there is a temporal issue. Since this study was an institutional-based study, it's difficult to generalize to the general population. In

addition, this study doesn't incorporate qualitative methods. If the study had used qualitative methods, it could have provided more detailed in-depth information to explain complex issues such as; behavior, and attitudinal factors which may not be adequately addressed by the quantitative method. We suggest qualitative studies be conducted in the future to provide an additional layer of analysis.

## Conclusion

Overall, there is a good level of prevention practice among pregnant women toward WHO-recommended COVID-19 prevention measures, but it needs improvement. In order to improve prevention practices among pregnant women, we recommend Harari Regional Health Bureau should provide repeated, targeted, and tailored information to pregnant women and the community at large through different media. In addition, by integrating different stakeholders, including health, educational, and religious institutions, to strengthen the recommended WHO prevention practice.

## Supporting information

**S1 Checklist. STROBE statement—Checklist of items that should be included in reports of *cross-sectional studies.***
(DOCX)

**S1 File. Annex: Questionnaire for assessing prevention practice and determinants of who recommended Covid-19 prevention measures among pregnant women attending antenatal care during the third wave of COVID-19 in Eastern Ethiopia, 2021.**
(DOCX)

**S2 File.**
(DOCX)

## Acknowledgments

First of all, we would like to express our thanks to the Almighty God. We would also like to extend our gratitude to the study subjects who participated in the study, Mrs. Ikram Mohammed and Dr. Abebe Desalegn for editing the language and grammatical flow throughout the manuscript, and to Harar Health Science College for providing us the opportunity to conduct the study.

## Author Contributions

**Conceptualization:** Astawus Alemayehu, Mohammed Yusuf, Abebaw Demissie, Yasin Abdullahi, Lemessa Oljira.

**Data curation:** Astawus Alemayehu, Mohammed Yusuf, Abebaw Demissie, Yasin Abdullahi, Lemessa Oljira.

**Formal analysis:** Astawus Alemayehu, Mohammed Yusuf, Abebaw Demissie, Yasin Abdullahi, Lemessa Oljira.

**Funding acquisition:** Astawus Alemayehu, Mohammed Yusuf, Abebaw Demissie, Yasin Abdullahi, Lemessa Oljira.

**Investigation:** Astawus Alemayehu, Mohammed Yusuf, Abebaw Demissie, Yasin Abdullahi, Lemessa Oljira.

**Methodology:** Astawus Alemayehu, Mohammed Yusuf, Abebaw Demissie, Yasin Abdullahi, Lemessa Oljira, Nega Assefa.

**Project administration:** Astawus Alemayehu, Mohammed Yusuf, Abebaw Demissie, Yasin Abdullahi, Lemessa Oljira.

**Resources:** Astawus Alemayehu, Mohammed Yusuf, Abebaw Demissie, Yasin Abdullahi, Lemessa Oljira.

**Software:** Astawus Alemayehu, Mohammed Yusuf, Abebaw Demissie, Yasin Abdullahi, Lemessa Oljira.

**Supervision:** Astawus Alemayehu, Mohammed Yusuf, Abebaw Demissie, Yasin Abdullahi, Lemessa Oljira.

**Validation:** Astawus Alemayehu, Mohammed Yusuf, Abebaw Demissie, Yasin Abdullahi, Lemessa Oljira, Nega Assefa.

**Visualization:** Astawus Alemayehu, Mohammed Yusuf, Abebaw Demissie, Yasin Abdullahi, Lemessa Oljira, Nega Assefa.

**Writing – original draft:** Astawus Alemayehu, Mohammed Yusuf, Abebaw Demissie, Yasin Abdullahi, Lemessa Oljira.

**Writing – review & editing:** Astawus Alemayehu, Mohammed Yusuf, Abebaw Demissie, Yasin Abdullahi, Lemessa Oljira, Nega Assefa.

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
