## [Decision Letter · Decision Letter 0]

1 Aug 2022

PONE-D-22-07011Prevention practice and determinants of WHO recommended COVID-19 prevention measures among pregnant women attending antenatal care during the third wave of COVID-19 in eastern Ethiopia, 2021: an institutional-based cross-sectional studyPLOS ONE

Dear Dr. Alemayehu,

Thank you for submitting your manuscript to PLOS ONE. After careful consideration, we feel that it has merit but does not fully meet PLOS ONE’s publication criteria as it currently stands. Therefore, we invite you to submit a revised version of the manuscript that addresses the points raised during the review process. We apologize for the delay incurred on your submission. 

The manuscript has been evaluated by two reviewers, and their comments are available below. They mainly request additional information on methodological aspects of the study and presentation/discussion of the results. Could you please revise the manuscript to carefully address the concerns raised?

We look forward to receiving your revised manuscript.

Kind regards,

Dario Ummarino, PhD

Senior Editor

PLOS ONE

Journal Requirements:

3. In the ethics statement in the Methods, you have specified that verbal consent was obtained. Please provide additional details regarding how this consent was documented and witnessed, and state whether this was approved by the IRB

We would also like to extend our gratitude to the study subjects who participated in the study and to Harar Health Science College for funding this study. 

However, funding information should not appear in the Acknowledgments section or other areas of your manuscript. We will only publish funding information present in the Funding Statement section of the online submission form. 

This study was funded by Harar Health Science College. funders plays a role in the study design, data collection and analysis, decision to publish, or preparation of the manuscript.

Reviewers' comments:

Reviewer's Responses to Questions

**Comments to the Author**

1. Is the manuscript technically sound, and do the data support the conclusions?

Reviewer #1: Partly

Reviewer #2: Yes

2. Has the statistical analysis been performed appropriately and rigorously? 

Reviewer #1: Yes

Reviewer #2: Yes

3. Have the authors made all data underlying the findings in their manuscript fully available?

Reviewer #1: Yes

Reviewer #2: No

4. Is the manuscript presented in an intelligible fashion and written in standard English?

Reviewer #1: No

Reviewer #2: Yes

5. Review Comments to the Author

Reviewer #1: Comments to the Authors

This manuscript described “Prevention practice and determinants of WHO recommended COVID-19 prevention measures among pregnant women attending antenatal care during the third wave of COVID-19 in eastern Ethiopia, 2021: an institutional-based cross-sectional study” which is an important issue to assess the practice and predictors of COVID-19 prevention measures among more risky groups pregnant women particularly in the study area. Despite the interesting scope of the research, the manuscript needs minor revision by considering the comments written below.

Title: The title is appropriate, but needs modification try to make SMART (it’s 30 words)

Abstract

A. Result: Please what is your outcome of interest? Good practice Vs Poor practice or WHO recommended COVID-19 preventive measures practice? Please include all significant predictors of practice of COVID-19 preventive measures

B. Conclusion and recommendation: Make it short and correlate with the findings of the study.

Introduction

1. What makes your study differ from others; because your study also assessed level of practice towards COVID-19 preventive measures?

2. Please add recent study like…. Compliance of COVID-19 preventive measures among pregnant women….in Ethiopia

Methods

1. Is this study conducted at all 14 health facilities? If so, indicate the study was conducted at all 14 health facilities in Harari region ….. Under study setting section.

2. Eligibility Criteria: ……. and had the willingness to participate in the study were included. Do you think that you have conducted with correct sampling technique? …. This indicated that those pregnant women who were selected using the sampling technique and who were refused to participate were excluded in your study. That’s wrong; there is selection bias.

3. Why determined your sample size with 50%? Why not used studies conducted in Southern Ethiopia, Debre Berhan ….?

4. Write your Kth interval

5. Operational definitions and measurements: put your reference for your operational definition and write how you value respondents’ answer ‘‘for all knowledge, and practice questions participants who answered the “correct answer” or “Yes” were earned a “1 score”, whereas participants who gave the “wrong answer” or “No” were earned a “0 score” separately.

Result

A. Please write all identified factors under section ‘‘Factors associated with WHO recommended COVID-19 Prevention practice’’

B. L196: The level of prevention practice among pregnant women was 61.6%… It’s not clear; please indicate the level of practice as Good Vs Poor practice???

Discussion

A. The discussion substantially needs rewriting; clear and logical justification

B. Please use recent studies to compare and contrast…..

Conclusion:

- Needs to be more punchy and to the point/ findings.

Reviewer #2: Abstract

Well written

Background

Lines 61-65: Could the authors kindly update the statistics!

Lines 82-86: It would be great to compare the severity of COVID-19 statistics for pregnant women in African during the first and second waves!

Methods

Lines 167-172: What informed the authors decision to use the mean as a cut off points for good and poor knowledge/practice? That has to been backed with scientific evidence.

Results

Lines 208-217: Kindly report all odds ratios with their decimal points as in table 4.

Lines 216-217: The interpretation is not accurate. It is 76% lower odds BUT NOT 76% times……….

Authors should be consistent with the use of decimal points on the table and throughout the manuscript. They should either stick to one or two decimal points but not use both.

Discussion

Lines 250-252: Provide a citation to back the speculation.

Lines 258-259: Provide a citation to back the speculation.

Lines 262-263: The interpretation is incorrect. Kindly refer to table 4 to write the right interpretation.

Lines 263-265: Provide a citation to back the speculation.

Lines 267-271: Provide citations to back the speculations.

Since this was an institutional-based cross-sectional study, it can only be generalized as such. This should be stated as one of the weaknesses of the study.

6. PLOS authors have the option to publish the peer review history of their article (what does this mean?). If published, this will include your full peer review and any attached files.

Reviewer #1: No

Reviewer #2: No

---

## [Author Response · Author response to Decision Letter 0]

21 Nov 2022

attached as "Response to Reviewer" file doc.

---

## [Decision Letter · Decision Letter 1]

20 Dec 2022

PONE-D-22-07011R1Prevention practice and determinants of WHO recommended COVID-19 prevention measures among pregnant women attending antenatal care during the third wave of COVID-19 in eastern Ethiopia, 2021PLOS ONE

Dear Astawus Alemayehu,

Thank you for submitting your manuscript to PLOS ONE. After careful consideration, we feel that it has merit but does not fully meet PLOS ONE’s publication criteria as it currently stands. Therefore, we invite you to submit a revised version of the manuscript that addresses the points raised during the review process.

ACADEMIC EDITOR:I agree with the reviewers that the manuscript needs further revisionOverall, the manuscript needs language editing. Please obtain language editing assistance before you submit the revised manuscript.Your research paper’s title is 28 words long. If possible, shorten it.In the financial disclosure, the authors state that Harar Health Sciences College played a role in the study design, data collection and analysis, decision to publish, or manuscript preparation. What were the interests and positions of the funder in the study? How did this impact the data collection, analysis, and interpretation? Whose views are reflected in the results? Were there any potential conflicts? How have you managed them? In the revised manuscript, in the Financial Disclosure section, discuss this and how you managed it. ==============================

We look forward to receiving your revised manuscript.

Kind regards,

Bereket Yakob, Ph.D.

Academic Editor

PLOS ONE

Journal Requirements:

Additional Editor Comments:

Reviewers' comments:

Reviewer's Responses to Questions

**Comments to the Author**

1. If the authors have adequately addressed your comments raised in a previous round of review and you feel that this manuscript is now acceptable for publication, you may indicate that here to bypass the “Comments to the Author” section, enter your conflict of interest statement in the “Confidential to Editor” section, and submit your "Accept" recommendation.

Reviewer #1: All comments have been addressed

Reviewer #2: (No Response)

2. Is the manuscript technically sound, and do the data support the conclusions?

Reviewer #1: Yes

Reviewer #2: Partly

3. Has the statistical analysis been performed appropriately and rigorously? 

Reviewer #1: Yes

Reviewer #2: Yes

4. Have the authors made all data underlying the findings in their manuscript fully available?

Reviewer #1: Yes

Reviewer #2: (No Response)

5. Is the manuscript presented in an intelligible fashion and written in standard English?

Reviewer #1: (No Response)

Reviewer #2: No

6. Review Comments to the Author

Reviewer #1: Thank you for your response.

More or less comments were addressed. Please, your topic needs modification to make SMART.

Also, rewrite and cite your operational definitions.

Reviewer #2: Introduction

The authors insist on using outdated statistics for the introduction. This is not an acceptable practice of international standards in the research community. It is important for the authors to note that they are writing the manuscript to inform current and future policy, but not past policy. It is therefore most appropriate to use current statistics. They should kindly address my earlier two comments on the introduction.

Methods

Authors should kindly cite the scientific evidence used for the mean cut off point in the appropriate section of the manuscript. It is not enough to only write that on the response page.

Results

Table 4 still contain odds ratios with different decimal points. Please make sure all decimal points are uniform (i.e. one decimal point or two decimal points for all).

Discussion

The authors claim that their findings are novel hence they cannot cite them. I do not agree that these novel findings. But even if these are novel findings, the reasons behind those findings MUST be accompanied by reasonable scientific evidence. So I urge the authors to kindly do the needful and cite appropriately.

7. PLOS authors have the option to publish the peer review history of their article (what does this mean?). If published, this will include your full peer review and any attached files.

Reviewer #1: **Yes: **Mulualem Silesh

Reviewer #2: **Yes: **Maxwell Tii Kumbeni

---

## [Author Response · Author response to Decision Letter 1]

10 Feb 2023

Attached as a document named 'Response to Reviewers".

---

## [Editor Report · Decision Letter 2]

14 Feb 2023

PONE-D-22-07011R2Determinants of WHO recommended COVID-19 prevention measures among pregnant women attending antenatal care during the third wave of COVID-19 in eastern Ethiopia, 2021PLOS ONE

Dear Dr. Astawus,

Thank you for submitting your manuscript to PLOS ONE. After careful consideration, we feel that it has merit but does not fully meet PLOS ONE’s publication criteria as it currently stands. Therefore, we invite you to submit a revised version of the manuscript that addresses the points raised during the review process.

We look forward to receiving your revised manuscript.

Kind regards,

Bereket Yakob, Ph.D.

Academic Editor

PLOS ONE

Journal Requirements:

Additional Editor Comments:

Dear Dr. Astawus,

Thanks for submitting the revised manuscript! Before I make the final decision regarding your manuscript, I request that you address the following.

1. Your responses to the Financial Disclosure query was inadequate. Based on the statement you provided in the financial disclosure, it needs to be clarified whether the authors' views, the institutional views, or both were reflected in the manuscript. Suppose the institution provided funds for the study and did not influence methods and participant selection, analysis, interpretation, and conclusion; it can be assumed that the views are of the authors and that there was no conflict of interest in the study. However, you stated that the funder, the institution in your particular case, played in all critical phases of the study – which shows it had a substantial influence in the study, leading to the assumption that there was a conflict of interest. If the statement was inserted by mistake, please revise it or state the institution's positions (including the formal statements, citations, or policy documents) about the study subject and how it impacted the findings, interpretations, and conclusion.

2. You added a new author to the manuscript. Please provide justifications and how the new author meets the authorship criteria.

3. Page # I agree with Reviewer 2 that you should discuss the findings/ relationships between family size and occupation of pregnant women by leaving out the novelty concern, which is debatable. Your study had only 6 people in the “good” category, which is insufficient to make such a bold statement. Since you had a small sample size in that category, it could have been merged with the other category (5-9 people). Besides, Page# 13, lines 230-232, is unclear – the language needs improvement. Please rephrase it to convey the message.

4. In the abstract, methods section, discuss/ describe how you measured the outcome variable

5. Your manuscript needs proofreading and grammar check.
---

## [Author Response · Author response to Decision Letter 2]

30 Mar 2023

we have attached a doc file as "Response to Reviewers"

---

## [Editor Report · Decision Letter 3]

5 Apr 2023

Determinants of WHO recommended COVID-19 prevention measures among pregnant women attending antenatal care during the third wave of COVID-19 in eastern Ethiopia, 2021

PONE-D-22-07011R3

Dear Astawus Alemayehu,

We’re pleased to inform you that your manuscript has been judged scientifically suitable for publication and will be formally accepted for publication once it meets all outstanding technical requirements.

Kind regards,

Bereket Yakob, Ph.D.

Academic Editor

PLOS ONE
---

## [Editor Report · Acceptance letter]

17 May 2023

PONE-D-22-07011R3 

Determinants of WHO recommended COVID-19 prevention measures among pregnant women attending antenatal care during the third wave of COVID-19 in eastern Ethiopia, 2021 

Dear Dr. Alemayehu:

I'm pleased to inform you that your manuscript has been deemed suitable for publication in PLOS ONE. Congratulations! Your manuscript is now with our production department. 

Kind regards, 

on behalf of

Dr. Bereket Yakob 

Academic Editor

PLOS ONE